# Abundance and Spatial Distribution of Aerobic Anoxygenic Phototrophic Bacteria in Tama River, Japan

**Yuki Sato-Takabe [1,2,\*]**, **Setsuko Hirose [1]**, **Tomoyuki Hori [2]** and **Satoshi Hanada [1,\*]**

1   Department of Biological Sciences, Tokyo Metropolitan University, 1-1 Minami-Osawa, Hachioji, Tokyo 192-0397, Japan; s-hirose@tmu.ac.jp
2   National Institute of Advanced Industrial Science and Technology, 16-1 Onogawa, Tsukuba 305-8569, Japan; hori-tomo@aist.go.jp
\*   Correspondence: yuki-takabe@aist.go.jp (Y.S.-T.); satohana@tmu.ac.jp (S.H.); Tel.: +81-29-861-8769 (Y.S.-T.)

**Abstract:** Aerobic anoxygenic phototrophic bacteria (AAnPB) are widely distributed and regarded as key players driving the carbon cycle in surface water of global oceans, coastal and estuary areas and in other freshwater environments (e.g., ponds and lakes). However, the abundance and spatial distribution of AAnPB in rivers is much less well-known. Here we investigated the variation of the absolute cell abundances of the total bacteria, AAnPB and cyanobacteria, at four different sites in Tama River, Japan, and the spatial distribution (i.e., free-living or particle-attached existence form) of AAnPB at two out of the four sites using infra-red epifluorescence microscopy. Free-living cell abundances for the total bacteria, AAnPB and cyanobacteria were $1.6–3 \times 10^5$, $1.5–4.4 \times 10^4$ and $<3.2 \times 10^4$ cells mL$^{-1}$, respectively. The free-living AAnPB accounted for 6.1%–19.6% of the total bacterial abundance in the river. The peaks of the AAnPB and cyanobacteria abundances were found at the same site, suggesting that the AAnPB could potentially coexist with cyanobacteria. Meanwhile, the particle-attached AAnPB were observed at the two sites of the river, accounting for 52.2% of the total bacteria abundance in the particle. Our results showed the existence and aggregation form of AAnPB in the riverine environment.

**Keywords:** aerobic anoxygenic phototrophic bacteria; river; bacteriochlorophyll

## 1. Introduction

Aerobic anoxygenic phototrophic bacteria (AAnPB) have bacteriochlorophyll (BChl) *a* and utilize the phototrophy and heterotrophy for energy acquisition [1]. The AAnPB primarily utilize organic matter as a carbon source and can use light as an additional source of energy. This additional energy production using photosynthetic reaction is expected to lead their effective growth and survival in natural environments [2,3]. The AAnPB are taxonomically distributed in the phyla of the *Alpha-, Beta-, Gamma-proteobacteria* and *Gemmatinomadetes* [4]. AAnPB have a wide distribution in the ocean [3,5,6] in which the proportion of AAnPB relative to the total bacteria reaches as high as 24% [5,7]. The main controlling factor of the AAnPB abundance was reported to be the organic matter concentration [4]. The wide range of the concentration would affect the AAnPB abundance levels in the ocean. Some previous reports showed that AAnPB contributed profoundly to the carbon stock and cycle in the ocean [3,5,7]. In lakes and ponds, AAnPB were also frequently observed albeit with the wide variation in relative abundance (i.e., 1%–37% of the total bacteria) [8–15] and could make the contribution in the freshwater systems as well. However, in contrast to those in marine environments, the abundance and spatial distribution of AAnPB in the freshwater environments, especially rivers, remain poorly understood. The first discovery of AAnPB has been made in the ocean [2,3] and thereafter researches have been conducted intensively on the same setting. The important role of AAnPB in the ocean carbon cycle has been well recognized [4]. Recently, the research scope of AAnPB has been expanded to freshwater



environments. Accumulating the basic data on the riverine AAnPB is of particular importance to have a better understanding of the microbial guild ecology.

Concerning the existence form of the AAnPB cells, Kolber et al., suggested that almost all the AAnPB cells in the tropical Pacific were free living [2]. Meanwhile, Lami et al., reported that the particle-attached AAnPB were dominant in the coastal area with high inflows of water from rivers [16]. Waidner & Kirchman also reported large proportions of particle-attached AAnPB in the Delaware estuary [17]. Findings from these previous studies suggested the universal existence of particle-attached AAnPB in rivers with rapid water flow [16,17]. However, little is known about the abundance ratio of AAnPB in the microbial aggregates in the river environments.

The objective of this study was to demonstrate the existence of the AAnPB and to clarify their existence form in Tama River, Japan, by investigating the absolute abundance of the total bacteria, AAnPB and cyanobacteria at four different sites in the middle parts of the river and the abundance ratios of particle-attached AAnPB to the total bacteria in the aggregates at two out of the four sites of the river, using infra-red epifluorescence microscopy.

## 2. Materials and Methods

### 2.1. Sampling Sites and the Reported Geochemical Parameters

Tama River has a length of 138 km and drains into Tokyo Bay after flowing from west to east over almost the entire length of Tokyo, Japan [18]. It is the largest river in Tokyo and has a basin area of 1240 $km^2$ that extends into neighbor prefectures. This study was conducted on the main upper-mid portion of Tama River at the four sampling sites. One site was located in the upper stream (i.e., site OM located in Oume city) and three sites were located in the middle stream (i.e., sites OZ, FS and KM located in Ozaku, Fussa and Komiya cities, respectively). Locations' information is 35.79 N, 139.25 E, 35.77 N, 139.29 E, 35.74 N, 139.32 E and 35.69 N, 139.37 E, at the sites OM (Oume), OZ (Ozaku), FS (Fussa) and KM (Komiya), respectively (Figure 1). The distances from the sampling sites OM, OZ, FS and KM to estuary into Tokyo Bay are approximately 65, 50, 45 and 40 km, respectively. Samples were taken at the sites OM, OZ, FS and KM in June 2017. The sampling was conducted one month before (i.e., in May 2017) at the site OM, and the obtained sample is designated as "OM0". Water samples were collected into the sterilized 50-mL centrifuge tubes that was rinsed three times with small amounts of water from the collection sites. Environmental parameters (i.e., water temperature and pH) were measured using a field equipment (LAQUAtwin; HORIBA, Kyoto, Japan). Samples were refrigerated during the transfer to the laboratory, fixed with neutral formaldehyde filtered by 0.2-μm pore-size filters at the final concentration of 1.0% in 10 mL aliquots within 6 h after sampling, held overnight at 4 °C in the dark, and filtered onto a Nucleopore black polycarbonate membrane filters (0.2-μm pore-size) under gentle vacuum (≤20 cm Hg).

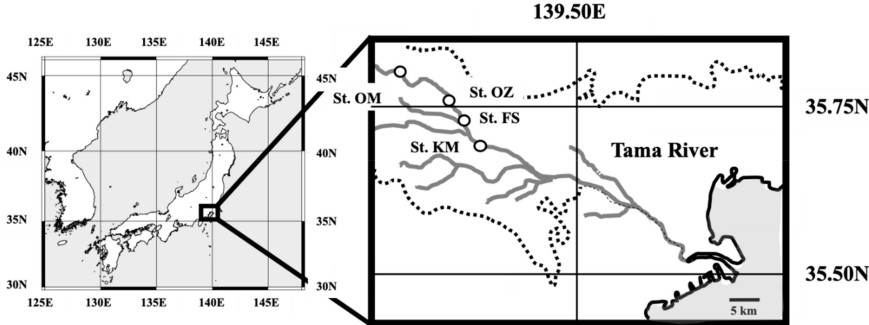

**Figure 1.** Map of the sampling sites in the present study. The left map indicates the map of Japan with the latitude and longitude, and the right map enclosed by bold square represents the study area. In the right map, the stations OM, OZ, FS and KM are located in Oume, Ozaku, Fussa and Komiya cities, respectively. The gray and thick line indicates Tama River. The solid and dot lines indicate the coastal line and prefecture's border, respectively. Scale bar indicates 5 km.

### 2.2. Infra-Red Epifluorescence Microscopy

The cell collection filters were dried, stained with 4′,6-diamidino-2-phenylindole (DAPI) at 1 µg mL$^{-1}$, and then mounted in a 3:1 ratio of Citiflour AF1 (Citifluor Ltd., London, United Kingdom) to Vectashield (Vector Labs, Burlingame, CA, USA). Bacterial cells were enumerated on images taken on an ECLIPSE E600 epifluorescence microscope (Nikon, Tokyo, Japan) equipped with a xenon lamp and a Photometrics CH-250 cooled, slow scan, infrared (IR)-sensitive CCD camera (Hamamatsu DIGITAL CAMERA C11440 ORCA-Flash 4.0, Hamamatsu Photonics, Okayama, Japan) and connected to an imaging software, NIS-Elements D software (Nikon, Japan). The following three epifluorescence filter sets were used: (1) BChl *a* (excitation 350–550 nm, emission > 830 nm long pass, > 665 nm long pass dichroic); (2) Chlorophyll (Chl) *a* (excitation 445 ± 45 nm, emission > 715 nm long pass, > 520 nm long pass dichroic); and (3) DAPI (excitation 365 ± 10 nm, emission > 400 nm long pass, 400 nm long pass dichroic). Two filters were used for the filtration of the river water and the repetition was checked in both the filters. The cell-counting experiment with one filter per each sample was conducted. Eight sets of the counting data per one filter were obtained to check the reproducibility. Bacterial cells were distinguished from other eukaryotic cells, based on the cell size. Based on the bacterial morphology found in aquatic environments, the bacterial cell size was generally considered to be <1 µm [19]. The acquired images were saved and semi-manually analyzed with the aid of a NIS-Elements D software. Microscope fields were selected at random, except for a few instances in which large phytoplankton cells or detrital particles dominated the field of view. Filamentous bacteria were not counted in the present study.

Cells were distinguished between non-phototrophic bacteria (Total bacteria—cyanobacteria—AAnPB), cyanobacteria and AAnPB. Here, approximately <1 µm sized Chl *a*-containing organisms were mainly determined as cyanobacterial cells. More than 1 µm sized ones were not counted as cyanobacterial cell. Approximately <1 µm sized Chl *a*-containing organisms consist mainly of cyanobacteria but could consist partially of small green algae. Cells appearing on the DAPI filter set (800 ms exposure) were identified at the total bacteria stained by DAPI. Cells appearing on the Chl *a* filter set (800 ms exposure) were identified as Chl *a*-containing organisms. Cell appearing on the BChl *a* filter set with IR emission (>830 nm) with 800 ms exposure were identified as both AAnPB and Chl *a*-containing organisms. AAnPB were identified as cells visualized by both DAPI and IR fluorescence but not Chl *a* fluorescence. The scheme for the identification of AAnPB from the other cells is shown in Table 1.

**Table 1.** Scheme for the identification of AAnPB in epifluorescence microscope imaging.

| Organisms | DAPI | Chl *a* | BChl *a* + Chl *a* |
|---|---|---|---|
| Heterotrophs | + | - | - |
| Cyanobacteria | + | + | + |
| AAnPB | + | - | + |

The abundance of AAnPB cells in particle was not determined as an absolute cell count but was reported as the ratio (AAnPB%). The ratio of AAnPB to total bacteria (%) was calculated with the following equation:

$$\text{AAnPB\%} = (\text{AAnPB and cyanobacterial cells appearing on the BChl } a \text{ filter set} - \text{cyanobacterial cells appearing on the Chl } a \text{ filter set})/\text{DAPI-stained bacterial cells per unit area.} \quad (1)$$

## 3. Results and Discussion

### 3.1. Determination of Absolute Cell Abundances of the Total Bacteria, Cyanobacteria and AAnPB

Based on the geochemical parameters of Tama River reported by the Bureau of Environment, Tokyo Metropolitan Governments (http://www.kankyo.metro.tokyo.jp/index.html), the total nitrogen

concentration tended to increase and the dissolved oxygen (DO) and total phosphate concentration tended to decrease from upstream to downstream in May and June 2017 (Table 2).

**Table 2.** Geochemical parameters of the present sampling sites.

| Site | WT (°C) | pH | DO (mg L$^{-1}$) | TN (mg L$^{-1}$) | TP (mg L$^{-1}$) |
|---|---|---|---|---|---|
| OM0 | 16.8 | 8.32 | 11.0 | 0.48 | 0.009 |
| OM | 14.9 | 9.00 | 10.8 | 0.48 | 0.011 |
| OZ | 17.6 | 9.10 | N.D. | N.D. | N.D. |
| FS | 22.6 | 8.00 | 10.2 | 0.53 | 0.009 |
| KM | 22.9 | 8.50 | N.D. | N.D. | N.D. |

WT: Water Temperature, DO: Dissolved Oxygen, TN: Total Nitrogen, TP: Total Phosphate, N.D.: 'No Data'. DO, TN and TP were cited by monthly report by the Bureau of Environment, Tokyo Metropolitan Government (http://www.kankyo.metro.tokyo.jp).

In the present study, most of the samples (i.e., four [OM, OZ, FS and KM] out of the five samples) were collected in the same time in June. Only one sample (i.e., OM0) was collected a month earlier, i.e., in May. Compared between the data of OM0 and OM, there was no significant differences in the cell abundances of the total bacteria, AAnPB and cyanobacteria ($p > 0.1$, two-tailed *t*-test), confirming the reproducibility of the obtained data within one month.

The total bacterial abundances ranged from $1.6 \times 10^5$ to $4.3 \times 10^5$ cells mL$^{-1}$ at the sites OM, OZ, FS and KM (Figure 2a). The total bacterial abundance at the site KM located in the most downstream river part was significantly higher than those at the other sites OM, OZ and FS ($p < 0.01$, two-tailed *t*-test). The monthly report of geochemical parameters in June 2017 indicates that the total nitrogen concentration showed increasing trends and DO showed decreasing trends from up- to down-streams of the river. The total bacterial abundances would reflect the biological activities fueled by the nutrient and oxygen. The AAnPB were detected at all the four sites of Tama River, showing the existence of AAnPB in the river areas. The abundances of free-living AAnPB ranged from $1.5 \times 10^4$ to $4.4 \times 10^4$ cells mL$^{-1}$ at the 4 sites (Figure 2b). The abundances of cyanobacteria ranged from under the detection limit to $3.2 \times 10^4$ cells mL$^{-1}$ at the four sites (Figure 2c) and were the highest at the site FS like AAnPB. The free-living AAnPB accounted for 6.1%–19.6% of the total bacterial abundance (Figure 2d), which was within the ranges reported in the previous studies on lakes and ponds [8–15]. The cyanobacteria accounted for 0.8%–12.8% of the total bacteria (Figure 2e). The AAnPB abundance did not show the obvious increasing tendency from the upper to middle streams, which was different from the total bacterial abundance. The peak of the AAnPB abundances was found at the site FS. The peaks of the AAnPB and cyanobacteria abundances were found at the same site. The statistical analysis (i.e., two-tailed *t*-test) was conducted for the evaluation of the differences in the cell abundance data among sampling sites in the present study. Due to the broad ranges of the standard deviations, most of the differences in the cell count data among sampling sites were not significant. Nevertheless, these results suggested that AAnPB could potentially coexist with cyanobacteria.

It has been reported in previous studies that a positive correlation between the AAnPB abundance and Chl *a* concentration in marine environments [20–22]. Mašín et al. [23] also reported that AAnPB abundance was positively correlated with Chl *a* concentration in glacial lakes of northern Europe. In both the marine and glacial lakes that have different geochemical parameters (e.g., salinity and temperature), the positive relationship between AAnPB abundance and Chl *a* concentration was found, which partially supports our data on the existence of AAnPB and cyanobacteria in the river. Because our data was not enough to lead the conclusion of the positive correlation between the abundances of AAnPB and cyanobacteria, the distribution of these types of bacteria in the river should be investigated more intensively in future.

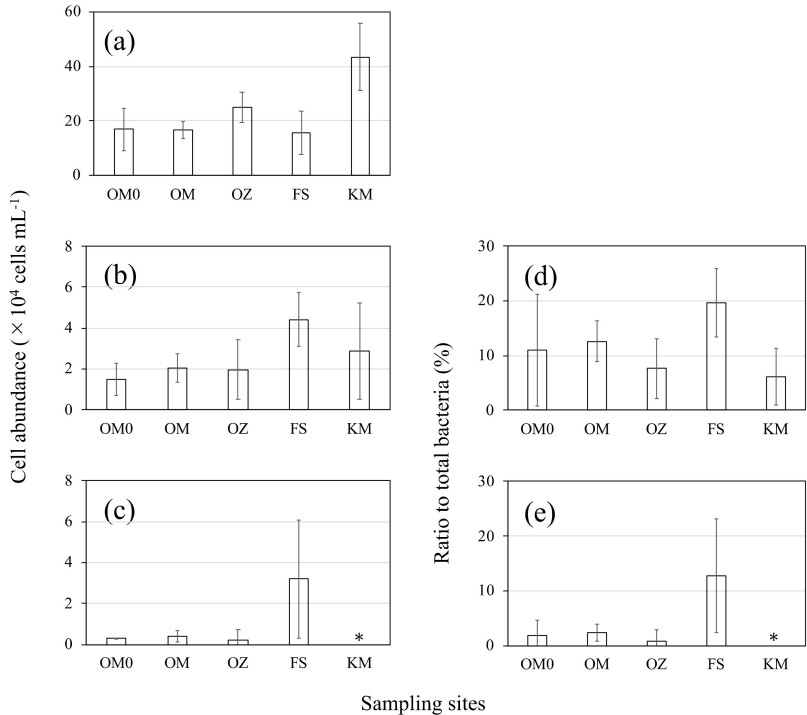

**Figure 2.** The absolute cell abundances of the total bacteria (**a**), AAnPB (**b**), cyanobacteria (**c**), the relative ratios of AAnPB (%) (**d**) and cyanobacterial (%) (**e**) in Tama River. Asterisks indicate 'under detection limit'.

### 3.2. Observation of the Particle-Attached AAnPB and Their Distribution Comparison with Free-Living AAnPB

Figure 3 shows the representative microscopic images obtained from infrared epifluorescence microscopy with different filters of free-living and particle-attached bacteria. At the sites OM and FS in the river, AAnPB had both the free-living and particle-attached existence forms.

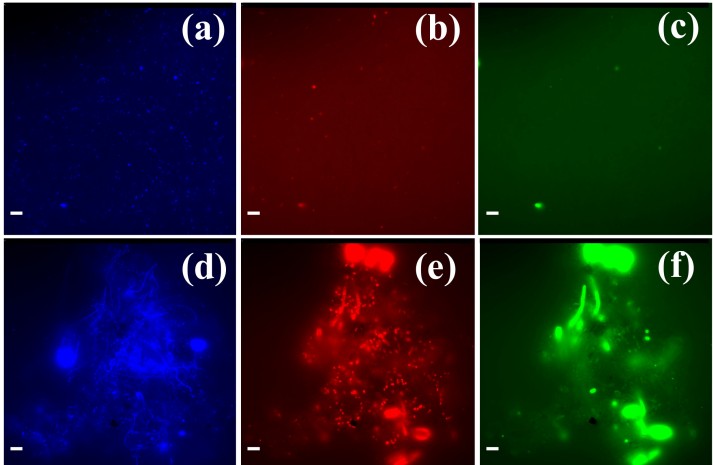

**Figure 3.** Representative images of free-living bacteria; (**a**) DAPI-stained for total cells (indicated in blue), (**b**) both Chl *a*-positive (cyanobacteria, autotrophic nanoflagellate and eukaryotic phytoplankton) and BChl *a*-positive (AAnPB) cells (indicated in red) and (**c**) Chl *a*-positive (cyanobacteria, autotrophic nanoflagellate and eukaryotic phytoplankton) cells (indicated in green). Those of particle-attached bacteria; (**d**) DAPI-stained for total cells (indicated in blue), (**e**) both Chl *a*-positive (cyanobacteria, autotrophic nanoflagellate and eukaryotic phytoplankton) and BChl *a*-positive (AAnPB) cells (indicated in red) and (**f**) Chl *a*-positive (cyanobacteria, autotrophic nanoflagellate and eukaryotic phytoplankton) cells (indicated in green). Each scale bar indicates 7 μm.

The ratio of AAnPB to the total bacteria in particles was 52.2 ± 13.9% ($n = 16$ in data at sites OM and FS). The ratios of particle-attached AAnPB were significantly higher than those of the free-living cells at these sites ($p < 0.01$ of the two-tailed *t*-test, Figure 4). Some of the previous studies reported the presence of the particle-attached AAnPB. Hirose et al. [24] detected the AAnPB species in the epilithic biofilm of Tama River. Most of first cultures of AAnPB species were isolated from the surface of seaweed [25]. This existence form of microbial cells, that is aggregation, was suggested to be preferential characteristics for AAnPB in natural environments. The cultured AAnPB species have been known to form aggregates and adhere to culture vessels [4]. In addition, Waidner & Kirchman [17] and Lami et al. [16] reported that a number of particle-attached AAnPB were found in estuary and coastal areas with the water input from river. The present and these previous studies suggested that the ubiquitous presence of the particle-attached AAnPB in rivers, as well as the estuary and coastal environments.

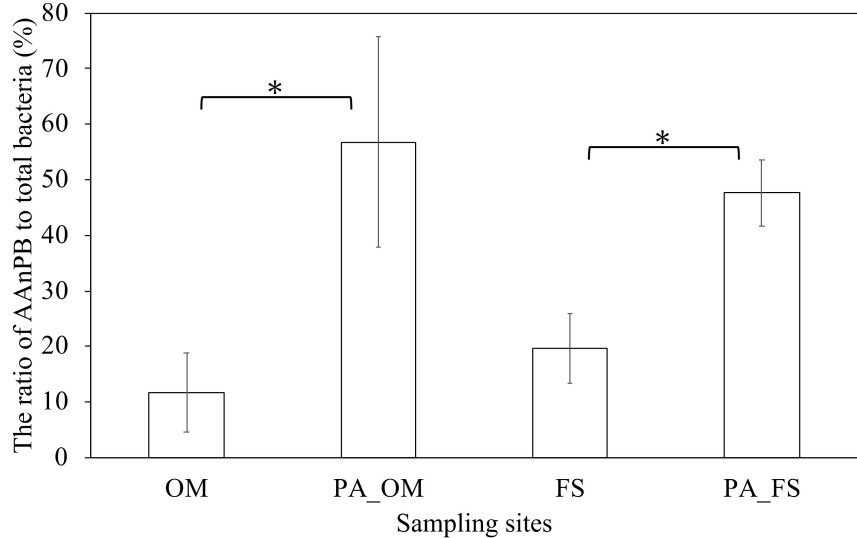

**Figure 4.** Free-living and particle-attached AAnPB ratio to the total bacteria (%) in Tama River. PA indicates 'particle-attached' and the others (unlabeled) are free-living. Asterisks represent $p < 0.01$ of the two-tailed *t*-test. Compared to AAnPB in lakes and ponds, the ecological knowledge of AAnPB in rivers is quite limited. Ruiz-Gonzáles et al. [26] described the distribution and abundance of AAnPB in the Ebro River located in the northern third of the Iberian Peninsula and indicated that the AAnPB abundances in the river changed seasonally from <1% in winter to 14% in autumn, suggesting their dramatic change depending on the seasonality. In the investigation of AAnPB in the ocean, there were wide variations of their abundances among various areas in the same season [4], which possibly resulted from the various geochemical parameters, for example, organic matter concentration and nutrient conditions and so on. In the present study, the absolute abundances of the free-living AAnPB in the river ranged from 6.1% to 19.6% of the total bacteria. The result apparently showed the variety of the AAnPB abundances in the different areas of the river but in the same time.

## 4. Conclusions

Our results indicated the abundance and spatial distribution of AAnPB in Tama River, Japan. The distribution pattern of the AAnPB was different from that of the total bacteria. The total bacterial abundances tended to increase with the nutrient concentration and the consumption of the dissolved oxygen, which it would reflect that the biological activity was promoted by the nutrient and oxygen. On the other hand, the AAnPB abundances did not show this increasing tendency, but the AAnPB appeared to potentially coexist with cyanobacteria, because the AAnPB abundance would more closely relate with the cyanobacterial abundance than the nutrient and oxygen concentrations. In addition, we showed the aggregation form of AAnPB in the river and the high relative ratio of AAnPB in the aggregates. Compared to the previous ecological knowledge of the AAnPB in rivers, our data firstly

showed the variety of their abundance in different areas of the river but in the same time. Because we focused on the specific area of the river and employed the limited number of the water samples, the distribution of the AAnPB in the various areas of the river (e.g., spring, mountain parts and estuary) should be investigated to clarify their ubiquitous existence and preferential existence form in the riverine environment.

**Author Contributions:** Conceptualization, Y.S.-T.; investigation, Y.S.-T. and S.H.; writing—original draft preparation, Y.S.-T.; writing—review & editing, Y.S.-T. and T.H.; funding acquisition, Y.S.-T., S.H. and T.H. All authors have read and agreed to the published version of the manuscript.

**Funding:** This research was funded by MEXT Grants-in-Aid for Young Scientists (80635839) for Y.S.-T. and also supported by a research grant from the Institute for Fermentation, Osaka.

**Acknowledgments:** This study was supported financially by MEXT Grants-in-Aid for Young Scientists (80635839) for Y.S.-T. and also supported by a research grant from the Institute for Fermentation, Osaka. We thank Yunosuke Nakamura for his helpful support in the microscopic imaging.

**Conflicts of Interest:** The authors declare no conflict of interest.

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
