# Peer review of "Abundance and Spatial Distribution of Aerobic Anoxygenic Phototrophic Bacteria in Tama River, Japan"

_water, doi:10.3390/w12010150_

Round 1

Reviewer 1 Report

This work was submitted as Communications. The paper reports abundance of aerobic anoxygenic phototrophs in five samples collected from Tama River, Japan. This is a very small dataset. The presentation of the findings is mostly descriptive. The methods are sound and the results are compatible with previous observations. The novel element is enumeration of aerobic anoxygenic phototrophs in the in the microbial aggregates, where they represent more then one half of all bacteria. The work is based solely on epifluorescence microscopy, other background data (temperature, chlorophyll, nutrients) are not presented. The samples were not collected during the same time. It is not clear why only middle part of the river was sampled. Why not also spring, mountain part, or estuary.

Author Response

Reviewer1

This work was submitted as Communications. The paper reports abundance of aerobic anoxygenic phototrophs in five samples collected from Tama River, Japan. This is a very small dataset. The presentation of the findings is mostly descriptive. The methods are sound and the results are compatible with previous observations.

> We deeply appreciate the positive view and precise suggestion of the reviewer 1 on our manuscript. The reviewer summarized our data and suggested the important improvement points in the present study. As the reviewer pointed it out, the presentation of our findings was descriptive. We have rewritten the descriptions carefully and overcome the weakness following to all the reviewers’ comments.

The novel element is enumeration of aerobic anoxygenic phototrophs in the in the microbial aggregates, where they represent more then one half of all bacteria.

> Thanks. As the reviewer indicated, clarifying the form and abundance ratio of AAnPB in the microbial aggregates is the significant scientific output in this study. We have made no change in the manuscript.

The work is based solely on epifluorescence microscopy, other background data (temperature, chlorophyll, nutrients) are not presented.

> We thank you for the useful suggestion. Temperature and pH of the river water were measured by ourselves and other geochemical parameters (i.e., the dissolved oxygen [DO], total nitrogen [TN] and total phosphate [TP] concentrations) were cited from the monthly report by the Bureau of Environment, Tokyo Metropolitan Government. We have added the new Table 2 showing these geochemical parameters in the revised manuscript (Line 136), as suggested also by the reviewer 2. However, we acknowledge that we do not have Chl a concentration data that is not included in the revised manuscript.

The samples were not collected during the same time.

> Most of the samples (i.e., four [OM, OZ, FS and KM] out of the five samples) were collected in the same time in June. Only one sample (i.e., OM0) was collected a month earlier, i.e., in May. Compared between the data of OM0 and OM, there was no significant differences in the cell abundances of the total bacteria, AAnPB and cyanobacteria (P > 0.1, two-tailed t-test). We have added the brief description in the revised manuscript (Line 145-149).

Line 140-144:

In the present study, most of the samples (i.e., four [OM, OZ, FS and KM] out of the five samples) were collected in the same time in June. Only one sample (i.e., OM0) was collected a month earlier, i.e., in May. Compared between the data of OM0 and OM, there was no significant differences in the cell abundances of the total bacteria, AAnPB and cyanobacteria (P > 0.1, two-tailed t-test), confirming the reproducibility of the obtained data within one month.

It is not clear why only middle part of the river was sampled. Why not also spring, mountain part, or estuary.

> It is true that our data were obtained from only middle parts of the river. In this study, we would like to emphasize the large variation of AAnPB abundances even in the specific and rather close parts of the river. As you suggested, targeting other sampling points, such as spring, mountain parts and estuary, is quite attractive and interesting. Although the investigation on these various areas is beyond the scope of the present study, these interesting topics have been included as future work in the revised manuscript (Line 235-237).

Line 235-237:

For the determination of the controlling factors of AAnPB abundance, it is necessary to clarify the functional guild dynamics by investigating in various environmental areas (e.g., spring, mountain parts and estuary).

Reviewer 2 Report

Dear Authors, 

I have read the manuscript with high interest. It is generally well written. Nevertheless, I have some questions and remarks that should be addressed before it is published. 

Please provide better justification of the research. You are claiming that the topic was not extensively studied to date. However I have been able to find some recent manuscripts on the subject - 10.15244/pjoes/76039, 10.1128/AEM.02116-17, 10.1093/femsec/fix065

I have provided several comments in the enclosed file. Please make sure that the aim is clearly presented. What is also a major point, please check whether you can improve the methodology description, particularly regarding statistics and the number of samples. 

Also the presentation of results could be improved. Please refer to the comment left in the file.

Best of luck.

Author Response

Reviewer2

I have read the manuscript with high interest. It is generally well written. Nevertheless, I have some questions and remarks that should be addressed before it is published. Please provide better justification of the research.

> We deeply appreciate that the reviewer 2 reviewed our manuscript with high interest and kindness and provided many useful and fruitful suggestions to improve our manuscript. All of the reviewer comments were thoroughly addressed and incorporated into the revised manuscript.

You are claiming that the topic was not extensively studied to date. However I have been able to find some recent manuscripts on the subject - 10.15244/pjoes/76039, 10.1128/AEM.02116-17, 10.1093/femsec/fix065

> We would acknowledge the insufficient reference of some recent articles in the manuscript. We have cited the references that you indicated (Line 42 and 157).

I have provided several comments in the enclosed file. Please make sure that the aim is clearly presented. What is also a major point, please check whether you can improve the methodology description, particularly regarding statistics and the number of samples.

Also the presentation of results could be improved. Please refer to the comment left in the file.

Best of luck.

Line 25 Living flocks or planktonic cells. Style?

> We would acknowledge the inappropriate usage of ‘style’ to represent the existence form of microbial cells. As the reviewer indicated, ‘free-living’ means planktonic, while ‘particle-attached’ indicates flock-living. We have omitted the word of ‘style’ and changed it to ‘existence form’ in the revised manuscript (Line 17-18).

Line 30 Why in this paper you use different acronym than in previous papers?

> Because we would believe that the acronym of ‘AAnPB’ is better expression than previous one. Generally, ‘AAP bacteria’ has been used as the abbreviation of aerobic anoxygenic phototrophic bacteria. Based on this way, anaerobic anoxygenic phototrophic bacteria could be also abbreviated as ‘AAP bacteria’, which is bit confusing. Thus, we argue that ‘AAnPB’ is suitable for aerobic anoxygenic phototrophic bacteria, whereas ‘AnAnPB’ is for anaerobic anoxygenic phototrophic bacteria. We have not made any change in revised manuscript.

Line 38-39 There are other and quite recent papers on this topic.

> We would acknowledge our improper citation in the manuscript. To accurately introduce the background of the present study, we have cited the new references below (Line 42 and 157), as you suggested.

The newly cited references:

・VojtÄ•ch Kasalický, Yonghui Zeng, Kasia Piwosz, Karel Šimek, Hana Kratochvilová, Michal Kobližek (2017) Aerobic anoxygenic photosynthesis is commonly present within the Genus Limnohabitans. Appl Environ Microbiol 84: e02116-17

・Qiang Li, Ang Song, Wenjie Peng, Zhenjiang Jin, Werner E. G., Müller and Xiaohong Wang (2017) Contribution of aerobic anoxygenic phototrophic bacteria to total organic carbon pool in aquatic system of subtropical karst catchments, Southwest China: evidence from hydrochemical and microbiological study. FEMS Microb Ecol 93: doi: 10.1093/femsec/fix065

・Yingying Tian, Xingqiang Wu, Qichao Zhou, Oscar Omondi Donde, Cuicui Tian, Chunbo Wang, Bing Feng, Bangding Xiao (2018) Plo J Environ Stud 27: 871-879.

Line 51-54 That is on the aim of the study. Please describe the objective.

> We deeply appreciate your useful advice to make our objective clearer. The paragraph has been revised as follow and presented in the revised manuscript (Line 57-61).

Line 57-61:

The objective in this study were to investigate the absolute abundance of the total bacteria, AAnPB and cyanobacteria at four different sites in the middle parts of Tama River, Japan, using infra-red epifluorescence microscopy and further to determine the abundance ratios of particle-attached AAnPB to the total bacteria in the particles at two out of the four sites of the river in order to characterize the spatial distribution preference of AAnPB.

Line 74-76 This is a result or a part of discussion. Shouldn’t be here.

> Thank you for the useful suggestion. The sentence has been moved to result part (Line 131-134).

Line 78~ How many filters? How many repetitions from each sampling site?

> We used two filters for the filtration of the river water and checked the repetition in both the filters. Then, we conducted the cell-counting experiment with one filter per each sample. Eight sets of the counting data per one filter were obtained to check the reproducibility. We have included the information in the revised manuscript (Line 98-100).

Line 98-100:

Two filters were used for the filtration of the river water and the repetition was checked in both the filters. The cell-counting experiment with one filter per each sample was conducted. Eight sets of the counting data per one filter were obtained to check the reproducibility.

Line 78~ What about statistics? Do you used any statistical method to analyze your results?

> According to the reviewer suggestion, we have done the two-tailed t-test as statistical analysis and added the obtained data and arguments. As a result of the statistical analysis, due to the broad ranges of standard deviations, most of the differences in the cell count data among sampling sites were not significant. Nevertheless, we could get the information on the trend of the differences. We have added the descriptions related to the statistics in the revised manuscript (Line 162-166).

Line 162-166:

The statistical analysis (i.e., two-tailed t-test) was conducted for the evaluation of the differences in the cell abundance data among sampling sites in the present study. Due to the broad ranges of the standard deviations, most of the differences in the cell count data among sampling sites were not significant. Nevertheless, the information concerning the trend of the differences were obtained.

Line 115-118 I would recommended to connect this data to the abundance data that you gathered. It would be good for the reception of the results if you indicated that in single figure.

> We thank you for the useful advice to connect the geochemical parameters to the abundance data of microorganisms. According to the reviewer suggestion, we have added the new Table 2 (shown below) to indicate the geochemical parameters (Line 136).

Table 2. Geochemical parameters of the present sampling

Line 158-163 Please make a separate paragraph for conclusion.

> Thanks. We have made the separated paragraph for conclusion (Line 231-241).

Figure 1 In my opinion you should show your graphs in two columns.

> According to the reviewer suggestion, we have reconstructed the previous Figure 1 and added the two-column graph as the new Figure 2 in the revised manuscript (Line 177). In this context, we have determined the relative ratio of cyanobacteria and added the data as Figure 2e.

Figure 2 Why the images are not in colour? It is difficult to judge on basis of presented figures.

> Because just one fluorescence was detected in each microscopic observation, we decided that the color-based imaging is not necessary in this case. We have not made any changes in the revised. Thanks.

References Very few current references. Only 3 not older than 5 years. Please brush up your reference list, especially that there are more recent papers on the topic.

> We would acknowledge our insufficient citation of recent articles in the manuscript. We have added the additional references in the revised manuscript (Line 42 and 157).

The newly cited references:

・VojtÄ•ch Kasalický, Yonghui Zeng, Kasia Piwosz, Karel Šimek, Hana Kratochvilová, Michal Kobližek (2017) Aerobic anoxygenic photosynthesis is commonly present within the Genus Limnohabitans. Appl Environ Microbiol 84: e02116-17

・Qiang Li, Ang Song, Wenjie Peng, Zhenjiang Jin, Werner E. G., Müller and Xiaohong Wang (2017) Contribution of aerobic anoxygenic phototrophic bacteria to total organic carbon pool in aquatic system of subtropical karst catchments, Southwest China: evidence from hydrochemical and microbiological study. FEMS Microb Ecol 93: doi: 10.1093/femsec/fix065

・Yingying Tian, Xingqiang Wu, Qichao Zhou, Oscar Omondi Donde, Cuicui Tian, Chunbo Wang, Bing Feng, Bangding Xiao (2018) Plo J Environ Stud 27: 871-879

Reviewer 3 Report

The authors analyse the abundance and spatial distribution of aerobic anoxygenic phototrophic bacteria in a river in Japan at various locations. Although this is a very interesting analysis, I believe major revisions of this manuscript are necessary before considering for publications. 

My main concern is related with the methodology and the conclusions the authors draw. The authors are concluding ecological impact, but this is not possible with this kind of data. Only cell abundance and spatial distribution has been measured instead of amplicon sequencing for instance. Even amplicon sequencing would not be enough for ecological relevance as this method also only allows to infer the taxonomic diversity, instead a metagenomics-based approach, idealy coupled with metatranscriptomics or metaproteomics should be used if broad ecological implications should be drawn from the habitat. Furthermore, the different numbers are not statistically tested, which is necessary to evaulate differences. And the graphics need some time to spent on.

And below my detailed review: 

abstract

l.14: how can ecology be assessed with these methods?
l.16: why were only AAnPB and cyanobacteria investigated in detail? amplicon sequencing gives an overview of the entire bacterial community.
l.16: numbers smaller than 20 should be written as words, please correct in entire manuscript.
l.22: the conclusions on the peaks are very speculative, given that the only indication for estimating dynamics are abundance values. it is an indication but this is not the right method to infer community dynamics.

introduction

l.30: reference missing
l.34: AAnPB: what are the taxonomic groups that belong to this group? this should be made more clearer.
l.37: wide variation in relative abundances. what might be reasons for that?
l.40: remain poorly understood. what is the reason for that?
l.44-50: this paragraph is written very vague and should show clearer and more concise what work has already be done.
l.51: abundance numbers cannot lead to ecological knowledge
l.54: 2 sites: are the two sites a subset of the other four sampling sites? this should be made clearer.

materials and methods

l.60: reference missing
l.63: map would help to make the coordinates and locations more clearer. then you can also move the abbreviations to the figure caption.
l.67: why the difference in sampling timing? also, a longitudinal study might help to resolve seasonal or daily variations. how is reproducibility ensured?
l.74: when were these parameters measured and reported? more information about the concentrations would be desirable, otherwise this is almost impossible to reproduce.
l.79: weird hyphenation
l.84: you should give more information about your computational infrastructure than just saying Windows PC
l.88: distiniguishment of bacterial cells. I highly doubt that size alone is a reasonable criterium. do you have any references that provide that this is correct and adequate? there are many bacterial cells that have different cell sizes.
l.94: this is not very reproducible.
l.97-98: why not counting them as cyanobacterial cells?
following lines: better make a table for how the cells were classified. this would be much easier to comprehend.
l.106: put formula as formula, not in the text.

results and discussion

l.112: 3.1 not 3.1.1
l.115: how much higher, significantly higher, where higher? this is not clear.
l.117: where can information about the concentrations be found? end of line: high: in relation to what?
l.127: what is the statistical evidence for the results?
end of paragraph: you are using diverse references, but you are generalizing their results. what is the effect of different environmental conditions, e.g. on glaciers?
l.136: what do the microscopy figures bring to the paper?
l.138: what is meant with style?
l.140: what is the statistical evidence of those results?
l.142: change Some of previous results to Some of the ...
l.143: reference order is wrong
l.144 and following: the key message from those references and this study are not clear, please rewrite and make it more concise.
l.150 new subsection, perhaps conclusions?
l.152: are there underlying factors beyond seasonailty?
l.157: actually you are not looking into the same season but on the same day. this is not enough to infer about seasonality.
l.159: based on the available data, it is very difficult to infer the particle-associated style. only one point in time was sampled, and abundance and spatial distribution are not enough for that.
l.161: outlook should be clearer, how can an extension of this study look like?

figure 1

you should not use excel for that but spend some more time to take a reasonable tool to create the barcharts. besides, all of them have a different y scale so the graphs (apart from (c) are not comparable). And there are too many breaks on the y axes.

figure 2

caption needs to be revised as it is right now quite confusing.

figure 3

also don't use excel but a proper tool. what do the colours represent? how are the differences statistically tested? are they significant (this also applies for figure 1).

Author Response

Reviewer3

The authors analyse the abundance and spatial distribution of aerobic anoxygenic phototrophic bacteria in a river in Japan at various locations. Although this is a very interesting analysis, I believe major revisions of this manuscript are necessary before considering for publications.

> We deeply appreciate the strict evaluation of the experimental procedure and data and further indication of possible publication in Water from the reviewer 3.

My main concern is related with the methodology and the conclusions the authors draw. The authors are concluding ecological impact, but this is not possible with this kind of data.

> We acknowledge our incorrect usage of ‘ecological impact’ albeit with the limited data sets. We have specified and expressed what the essential of the present study is. Accordingly, the conclusion of this study has now been toned down appropriately (Line 57-61 and 232-235) and we have omitted the strong words, e.g., ‘ecological impact’ and ‘ecology’ from the revised manuscript.

Only cell abundance and spatial distribution has been measured instead of amplicon sequencing for instance. Even amplicon sequencing would not be enough for ecological relevance as this method also only allows to infer the taxonomic diversity, instead a metagenomics-based approach, idealy coupled with metatranscriptomics or metaproteomics should be used if broad ecological implications should be drawn from the habitat.

> We deeply appreciate that you kindly suggest the valuable and interesting works. Because investigating whole microbial communities by the amplicon sequencing and meta-omics tools is beyond the scope of this study, we would like to keep these challenges as our future works for a better understanding of the diversity and function of AAnPB in natural environments.

 Furthermore, the different numbers are not statistically tested, which is necessary to evaulate differences. And the graphics need some time to spent on.

> According to the reviewer comment, we have done the two-tailed t-test as statistical analysis and added the obtained data and arguments. As a result of the statistical analysis, due to the broad ranges of standard deviations, most of the differences in the cell count data among sampling sites were not significant. Nevertheless, we could get the information concerning the trend of the differences. We have added the descriptions related to the statistics in the revised manuscript (Line 162-166).

Line 141-144:

The statistical analysis (i.e., two-tailed t-test) was conducted for the evaluation of the differences in the cell abundance data among sampling sites in the present study. Due to the broad ranges of the standard deviations, most of the differences in the cell count data among sampling sites were not significant. Nevertheless, the information concerning the trend of the differences were obtained.

And below my detailed review: abstract

l.14: how can ecology be assessed with these methods?

> We would acknowledge our incorrect usage of the strong word ‘ecology’. According to the comment, we have replaced it with ‘the abundance and spatial distribution’ (Line 14-15).

l.16: why were only AAnPB and cyanobacteria investigated in detail? amplicon sequencing gives an overview of the entire bacterial community.

> Because AAnPB are considered as one of the main functional bacterial groups driving the carbon cycle in water environments, such as ocean. Furthermore, The Chl a-containing cyanobacteria have been known to have the close relationship with AAnPB, as indicated in previous reports (Koblížek, 2015). Therefore, we investigated herein the abundance and spatial distribution of AAnPB with the special focus on its relationship with cyanobacteria. As mentioned above, examining the whole bacterial community is out of scope in this study. We have made no changes in the revised manuscript.

Koblížek M. Ecology of aerobic anoxygenic phototrophs in aquatic environments. FEMS Microbiol Ecol 2015, 39, 854–870.

l.16: numbers smaller than 20 should be written as words, please correct in entire manuscript.

> Thanks. Corrected throughout the manuscript (Line 16, 18, 67, 68, 98-100, 151, 154).

l.22: the conclusions on the peaks are very speculative, given that the only indication for estimating dynamics are abundance values. it is an indication but this is not the right method to infer community dynamics.

> We acknowledge our inadequate usage of the strong word ‘dynamics’. According to the reviewer comment, we have replaced it with ‘abundance’ in the revised manuscript (Line 22).

introduction

l.30: reference missing

> Thanks. We have added the reference, Beatty 2002 [1] (Line 32).

l.34: AAnPB: what are the taxonomic groups that belong to this group? this should be made more clearer.

> We have added the information about the taxonomic groups of AAnPB in the revised manuscript (Line 35-36).

Line 35-36:

The AAnPB are taxonomically distributed in the phyla of the Alpha-, Beta-, Gamma-proteobacteria and Gemmatinomadetes [4].

l.37: wide variation in relative abundances. what might be reasons for that?

> In the recent review about AAnPB (Koblížek 2015), the main controlling factor of the AAnPB abundance was the organic matter concentration. The wide range of organic matter concentration would reflect that of the AAnPB abundance in the ocean. However, we think that there should be the other controlling factors that have not been found yet.

We have added the following argument in the revised manuscript (Line 38-40):

The main controlling factor of the AAnPB abundance was reported to be the organic matter concentration [4]. The wide range of the concentration would reflect that of AAnPB abundance in the ocean.

l.40: remain poorly understood. what is the reason for that?

> In the research field of the AAnPB ecology, the first discovery of AAnPB has been made in the ocean and then researches have been conducted intensively on the same setting. The important role of AAnPB in the ocean carbon cycle has been well recognized. Recently, the research scope of AAnPB have been expanded to freshwater environments. We have added the related arguments in the revised manuscript (Line 45-48)

Line 45-48:

The first discovery of AAnPB has been made in the ocean [2,3] and thereafter researches have been conducted intensively on the same setting. The important role of AAnPB in the ocean carbon cycle has been well recognized [4]. Recently, the research scope of AAnPB have been expanded to freshwater environments.

l.44-50: this paragraph is written very vague and should show clearer and more concise what work has already be done.

> We acknowledge our vague explanation about the previous studies. For clarity, we have omitted the first sentence and revised the sentences carefully as shown below (Line 50-56):

Line 50-56:

Concerning the existence form of the AAnPB cells, Kolber et al., suggested that almost all the AAnPB cells in the tropical Pacific were free living [2]. Meanwhile, Lami et al., reported that the particle-attached AAnPB were dominant in the coastal area with high inflows of water from rivers [16]. Waidner & Kirchman also reported large proportions of particle-attached AAnPB in the Delaware estuary [17]. Findings from these previous studies suggested the universal existence of particle-attached AAnPB in rivers with rapid water flow [16,17]. However, little is known about the form and abundance ratio of AAnPB in the microbial aggregates in the river environments.

l.51: abundance numbers cannot lead to ecological knowledge

> We acknowledge our improper usage of the word ‘ecological knowledge’. Accordingly, we have omitted the related sentence.

l.54: 2 sites: are the two sites a subset of the other four sampling sites? this should be made clearer.

> Yes, they are. We have added the clear description (Line 57-61), as shown below.

Line 57-61:

The objective in this study were to investigate the absolute abundance of the total bacteria, AAnPB and cyanobacteria at four different sites in the middle parts of Tama River, Japan, using infra-red epifluorescence microscopy and further to determine the abundance ratios of particle-attached AAnPB to the total bacteria in the particles at two out of the four sites of the river in order to characterize the spatial distribution preference of AAnPB.

materials and methods

l.60: reference missing

> Thanks. We have added the reference (Line 65)

The cited reference:

・Okai M, Aoki H, Ishida M, Urano N. Antibiotic-resistance of fecal coliforms at the bottom of the Tama River, Tokyo. Biocontrol Sci 24: 173-178.

l.63: map would help to make the coordinates and locations more clearer. then you can also move the abbreviations to the figure caption.

> We agreed with your suggestion. We have added the map of sampling sites as Figure 1 (Line 81). Thank you.

Figure 1. Map of the sampling sites in the present study. The stations OM, OZ, FS and KM are located in Oume, Ozaku, Fussa and Komiya cities, respectively. Blue and thick line indicates Tama River. Solid and dot line indicate coastal line and prefecture’s border, respectively. Scale bar indicates 5 km.

l.67: why the difference in sampling timing? also, a longitudinal study might help to resolve seasonal or daily variations. how is reproducibility ensured?

> In this study, most of the samples (i.e., four [OM, OZ, FS and KM] out of the five samples) were collected in the same time in June. Only one sample (i.e., OM0) was collected a month earlier, i.e., in May. Compared between the data of OM0 and OM, there was no significant differences in the cell abundances of the total bacteria, AAnPB and cyanobacteria (P > 0.1, two-tailed t-test), confirming the reproducibility of the obtained data within one month. As the reviewer indicated, the longitudinal study should be beneficial for resolving the seasonal or daily variations, which is, however, beyond the scope of this study. We would like to highlight the large variation of AAnPB abundances even at the same time and in the specific and rather close parts of the river. We have added the related arguments in the revised manuscript (Line 140-144).

Line 140-144

In the present study, most of the samples (i.e., four [OM, OZ, FS and KM] out of the five samples) were collected in the same time in June. Only one sample (i.e., OM0) was collected a month earlier, i.e., in May. Compared between the data of OM0 and OM, there was no significant differences in the cell abundances of the total bacteria, AAnPB and cyanobacteria (P > 0.1, two-tailed t-test), confirming the reproducibility of the obtained data within one month.

l.74: when were these parameters measured and reported? more information about the concentrations would be desirable, otherwise this is almost impossible to reproduce.

> Temperature and pH of the river water were measured by ourselves and other geochemical parameters (i.e., the dissolved oxygen [DO], total nitrogen [TN] and total phosphate [TP] concentrations) were cited from the monthly report by the Bureau of Environment, Tokyo Metropolitan Government. We have added the new Table 2 showing these geochemical parameters in the revised manuscript (Line 136), as suggested also by the reviewer 2.

l.79: weird hyphenation

> Thanks. I have confirmed that 4',6-diamidino-2-phenylindole is the formal name of DAPI (Sierackie et al. 2016). Thus, we have made no change in the text.

Sieracki M.E., Glig J.C., Their E.C., Poulton N.J., Goericke R. Distribution of planktonic aerobic anoxygenic photoheterotrophic bacteria in the northwest Atlantic. Limnol Oceanogr 2006, 51, 38–44.

l.84: you should give more information about your computational infrastructure than just saying Windows PC

> We would acknowledge the incomplete information on the computational infrastructure that we employed. We have changed the words ‘Windows PC’ to ‘imaging software, NIS-Elements D software (Nikon, Japan) (Line 93-94).’

l.88: distiniguishment of bacterial cells. I highly doubt that size alone is a reasonable criterium. do you have any references that provide that this is correct and adequate? there are many bacterial cells that have different cell sizes.

> Thank you very much for the valuable comments. The reviewer concern is reasonable, because of the fact that bacterial cells have different cell sizes. However, we took into account the specific case in natural environments. Owing to the book chapter of Kirchman, 2008 about bacterial morphology in aquatic environments, generally natural bacterial cell size was considered to be <1 μm. We have added the reference and related arguments in the revised manuscript (Line 126-127).

Line 127-178

Based on the bacterial morphology found in aquatic environments, the bacterial cell size was generally considered to be <1 μm [19].

Kirchman D.L. Chapter 1. Introduction and overview, In 2nd edition of Microbial ecology of the ocean. Ed. Kirchman D.L. 2008, Wiley, pp 1-6.

l.94: this is not very reproducible.

> We would acknowledge our poor description. Accordingly, the sentence was omitted from the manuscript (Line 105).

l.97-98: why not counting them as cyanobacterial cells? following lines: better make a table for how the cells were classified. this would be much easier to comprehend.

> Thanks for very useful suggestion. Although the identification criteria were already described in the main text, according to the reviewer suggestion, we have made the new Table 1 to indicate the clear criteria and added it in the revised manuscript (Line 118).

Table 1. Scheme for the identification of AAnPB in epifluorescence microscope imaging

l.106: put formula as formula, not in the text.

> We acknowledge our incorrect presentation of the formula. The formula has been put as formula accordingly (Line 124-125). Thank you.

results and discussion

l.112: 3.1 not 3.1.1

> Corrected. Thanks.

l.115: how much higher, significantly higher, where higher? this is not clear.

> We totally agreed with your opinion. To claify the important point, in this revision, we have done the two-tailed t-test as statistical analysis. The result confirmed that the sentence ‘Moving from the upper stream (site OM) to the middle streams (sites OZ, FS and KM), the total bacterial abundances tended to be higher.’ was not correct, because the total bacterial abundances at the sites FS, OM and OZ were comparable and only the abundance at site KM was significantly higher than those at the other sites. We have revised the sentence carefully, as shown below.

Line 146-148:

The total bacterial abundance at the site KM located in the most down-stream river part was significantly higher than those at the other sites OM, OZ and FS (P < 0.01, two-tailed t-test).

l.117: where can information about the concentrations be found? end of line: high: in relation to what?

> As mentioned above, we have added the new Table 2 showing the geochemical parameters of the river in the revised manuscript (Line 136). After the careful check and statistical analysis data, we have deleted the word ‘high’ at the end of the line (Line 150-151).

Line 150-151:

The total bacterial abundances would reflect the biological activities fueled by the nutrient and oxygen concentrations.

l.127: what is the statistical evidence for the results?

> As a result of the statistical analysis additionally performed, it has been confirmed that most of the differences in the cell count data among sampling sites were not significant. Nevertheless, we could get the important information on the trend of the differences. We have shown the correspondence of the abundance peaks of the AAnPB and cyanobacteria.

end of paragraph: you are using diverse references, but you are generalizing their results. what is the effect of different environmental conditions, e.g. on glaciers?

> Thanks for the detailed and significant comment. According to the literature, the positive relationship between AAnPB and Chl a concentration was observed in both the marine and glacier lakes that have the different geochemical parameters. We have added the related argument (Line 170-172), as shown below.

Line 170-172:

In both the marine and glacial lakes that have different geochemical parameters (e.g., salinity and temperature), the positive relationship between AAnPB abundance and Chl a concentration was found, which partially supports our data on the distribution of AAnPB and cyanobacteria in the river.

l.136: what do the microscopy figures bring to the paper?

> We deeply appreciate the useful comment. We would like to show the aggregation form of AAnPB by the actual microscopic images, which is one of the novel points in this study. We have not made any change in the text.

l.138: what is meant with style?

> We acknowledge our inappropriate usage of the word ‘style’. We have replaced it with ‘existence form’ in the revised manuscript.

l.140: what is the statistical evidence of those results?

> According to the reviewer suggestion, we have done the two-tailed t-test as statistical analysis and confirmed the statistical significance of the obtained data. We have added the descriptions related to the statistics in the revised manuscript (Line 162-166).

Line 162-166:

The statistical analysis (i.e., two-tailed t-test) was conducted for the evaluation of the differences in the cell abundance data among sampling sites in the present study. Due to the broad ranges of the standard deviations, most of the differences in the cell count data among sampling sites were not significant. Nevertheless, the information concerning the trend of the differences were obtained.

l.142: change Some of previous results to Some of the ...

> Thanks. Corrected (Line 200).

l.143: reference order is wrong

> Thanks. Corrected (Line 201).

l.144 and following: the key message from those references and this study are not clear, please rewrite and make it more concise.

> We totally agreed with the reviewer opinion. Exactly, this sentence was poorly described and did not have the clear connection with our data. We have revised the sentence carefully (Line 202-203), as shown below:

Line 202-203:

This existence form of microbial cells, that is aggregation, was suggested to be preferential characteristics for AAnPB.

l.150 new subsection, perhaps conclusions?

> Thanks. We have added the new subsection for conclusions.

l.152: are there underlying factors beyond seasonailty?

> In the investigation of AAnPB in the ocean, there were wide variations of their abundances among various areas in the same season (Kiblizek, 2015). The emergence of the wide variations were possibly due to the geochemical parameters, for example, organic matter concentration, nutrient conditions and so on. We have described the arguments in the revised manuscript (Line 222-225).

Line 222-225:

In the investigation of AAnPB in the ocean, there were wide variations of their abundances among various areas in the same season [4]. The emergence of the wide variations was possibly due to the geochemical parameters, for example, organic matter concentration, nutrient conditions and so on.

l 157: actually you are not looking into the same season but4

l.157: actually you are not looking into the same season but on the same day. this is not enough to infer about seasonality.

> We acknowledge our improper usage of the word ‘season’. We have removed ‘season’ and just described the sampling timing, e.g., month (Line 228).

l.159: based on the available data, it is very difficult to infer the particle-associated style. only one point in time was sampled, and abundance and spatial distribution are not enough for that.

> Thank you for the strict evaluation to improve our manuscript quality. At first, we have omitted the word ‘style’ and changed it the word ‘existence form of microbial cells’. Secondly, although our data was obtained from almost one time point, we have additionally done the statistical test and showed the statistically significant differences of the AAnPB ratio between the free-living cells and particle-attached cells (Figure 4). We have made no change in the revised manuscript.

l.161: outlook should be clearer, how can an extension of this study look like?

> We deeply appreciate your constructive suggestion. We have revised the sentence carefully (Line 237-241), as shown below:

Line 237-241

Further clarification of the physiological characteristics of AAnPB is needed for assessing why AAnPB prefer the particle-attached existence form as the spatial distribution. As the future works, the diversity and function of AAnPB-dominated microbial communities should be investigated by, for instance, the amplicon sequencing, metatranscriptomics and metaproteomics.

figure 1

you should not use excel for that but spend some more time to take a reasonable tool to create the barcharts. besides, all of them have a different y scale so the graphs (apart from (c) are not comparable). And there are too many breaks on the y axes.

> Thank you very much for the useful comment. Corrected accordingly. We have reconstructed the previous figure 1 carefully and presented it as the new Figure 2.

figure 2

caption needs to be revised as it is right now quite confusing.

> Corrected. Thank you.

figure 3

also don't use excel but a proper tool. what do the colours represent? how are the differences statistically tested? are they significant (this also applies for figure 1).

> We deeply appreciate your kind suggestion to do the statistical test. As mentioned above, we have done the statistical test. The results have now been shown as the new Figure 4 in the revised manuscript (Line 212).

Figure 4. Free-living and particle-attached AAnPB ratio to total bacteria (%) in Tama River. PA indicates ‘particle-attached’ and the others (unlabeled) are free-living. * represents P < 0,01, two-tailed t-test.

Round 2

Reviewer 2 Report

Dear Authors,

thank you for getting back with the reviewed manuscript. 

All of my comments were addressed. Even though not all of them resulted in changes in the manuscript, the authors stated their reasoning behind some of the raised points. 

I am glad that statistics were incorporated to the manuscript. Unfortunately, I can't agree to the argument made by authors that the tendency is enough to draw a conclusion. You assume that what you see is true, but show no evidence to back it up. In that case you should find another way to verify the result. 

Similarly with fluorescence microscopy images. I would insist on using images in colour. I believe this would increase the transparency of these photographs. Even though only one colour may be there, one should be able to distinguish between different cells. In monochromatic version it is not possible. 

I think that these points should be addressed again in the second revision round to make the manuscript better and ready for publication. 

Best of luck

Author Response

Reviewer2

Dear Authors,

thank you for getting back with the reviewed manuscript.

All of my comments were addressed. Even though not all of them resulted in changes in the manuscript, the authors stated their reasoning behind some of the raised points.

> We deeply appreciate the kind review, again. In the second round of the review for our manuscript, the reviewer 2 has also carefully reviewed our manuscript. All the suggestions below would help us to improve the quality of our manuscript.

I am glad that statistics were incorporated to the manuscript. Unfortunately, I can't agree to the argument made by authors that the tendency is enough to draw a conclusion. You assume that what you see is true, but show no evidence to back it up. In that case you should find another way to verify the result.

>We fully agree with the comments. According to the reviewer comments, the arguments on the relationship between AAnPB and cyanobacteria have been toned down drastically. In this regard, we have omitted the conclusion about the relation of the AAnPB and cyanobacteria abundances and just described ‘the potential coexistence of these types of bacteria’ as the new message. In addition, the necessity of the future works to verify the relationship between AAnPB and cyanobacteria was added in the revised manuscript.

The revised sentences are shown below:

Line 22-23

, suggesting that the AAnPB abundance were closely related not to the total bacterial but to the cyanobacterial dynamics.

=>, suggesting that the AAnPB could potentially coexist with cyanobacteria.

Line 157-158

These results strongly suggested that the AAnPB abundance were closely related to the cyanobacterial abundances but not to the total bacterial abundance.

=> These results suggested that the AAnPB could potentially coexist with cyanobacteria.

Line 161-162

Nevertheless, the information concerning the trend of the differences were obtained.

=> Deleted.

Line 168-170

The results in the present and previous studies implied that AAnPB preferentially utilized organic matter that was derived, for instance, from Chl a-containing organisms (such as cyanobacteria and phytoplankton) in the river environments.

=> Because our data was not enough to lead the conclusion of the positive correlation between the abundances of AAnPB and cyanobacteria, the distribution of these types of bacteria in the river should be investigated more intensively in future.

Similarly with fluorescence microscopy images. I would insist on using images in colour. I believe this would increase the transparency of these photographs. Even though only one colour may be there, one should be able to distinguish between different cells. In monochromatic version it is not possible.

>We thank you for the useful advicefor the improvement of the fluorescence microscopy images. We have changed the images in color, according to the reviewer suggestion. In Figure 3, the DAPI-stained cells are indicated in blue (Figs. 3 a and d). The Chl a- and BChl a-positive cells are indicated in red (Figs. 3 b and e), while the Chl a-positive cells are indicated in green (Figs. 3 c and f).

The legend of Figure 3 was revised as below:

Line 183-191

Figure 3. Representative images of free-living bacteria (a) DAPI-stained for total cells (indicated in blue), (b) both Chl a-positive (cyanobacteria, autotrophic nanoflagellate and eukaryotic phytoplankton) and BChl a-positive (AAnPB) cells (indicated in red) and (c) Chl a-positive (cyanobacteria, autotrophic nanoflagellate and eukaryotic phytoplankton) cells (indicated in green). Those of particle-attached bacteria (d) DAPI-stained for total cells (indicated in blue), (e) both Chl a-positive (cyanobacteria, autotrophic nanoflagellate and eukaryotic phytoplankton) and BChl a-positive (AAnPB) cells (indicated in red) and (f) Chl a-positive (cyanobacteria, autotrophic nanoflagellate and eukaryotic phytoplankton) cells (indicated in green). Each scale bar indicates 100 pixels = 7 μm.

I think that these points should be addressed again in the second revision round to make the manuscript better and ready for publication.

Best of luck

>We believe that we have effectively addressed all the reviewer’s concern. We hope that the revised manuscript is now suitable for the publication in Water. Thanks.

Reviewer 3 Report

l.32 and 33: twice reference [1] l. 240: change to by amplicon sequencing

Author Response

Reviewer3

> We deeply appreciate your careful correction on our manuscript.

l.32 and 33: twice reference [1]

> Corrected. We have deleted the latter reference [1]. (Line 32)

240: change to by amplicon sequencing

> Corrected. In this context, we have omitted the words ’for instance’ in the sentence.

The revised sentences are shown below:

Line 234-246

As the future works, the diversity and function of AAnPB-dominated microbial communities should be investigated by, for instance, the amplicon sequencing, metatranscriptomics and metaproteomics.

=> As the future works, the diversity and function of AAnPB-dominated microbial communities should be investigated by amplicon sequencing, metatranscriptomics and metaproteomics.

Round 3

Reviewer 2 Report

Dear Authors,

thank you for the work you did to improve the manuscript. 

All my comments were answered. I will be happy to read the final manuscript in its published form. 

Best wishes

Author Response

Dear Reviewer 2

We are so happy that you have been satisfied with our revision.

We deeply appreciate your kind review, again.

Many thanks,

Yuki Sato-Takabe